# Growth, proportion, and distribution pattern of longleaf pine across southeastern forests and disturbance types: A change assessment for the period 1997-2018

**Santosh K. Ojha**[1]\*, **Luben D. Dimov**[2], **Wubishet Tadesse**[1]

**1** Department of Biological and Environmental Sciences, Alabama A&M University, Normal, AL, United States of America, **2** Rubenstein School of Environment and Natural Resources, University of Vermont, Burlington, VT, United States of America

\* santosh.ojha@aamu.edu

## Abstract

The long-term decline of longleaf pine-dominated forests has received considerable attention among land managers and conservation professionals in the last few decades. The objective of this study was to investigate the change in and the variation of the proportion, density, growth, and dominance of longleaf pine across the longleaf pine ecosystems for the 1997–2018 period. We used two sets of measurements of 1,432 plots from the Forest Inventory and Analysis (FIA) dataset covering the entire current longleaf pine range. The relationship between disturbances and longleaf pine basal area ratio and basal area growth were analyzed using linear mixed modeling. Change detection maps were produced using the Inverse Distance Weighted (IDW) interpolation method. The total basal area and above-ground biomass per hectare increased in 64% and 72%, but decreased in 30% and 28% of the study area, respectively, between the first and last inventory intervals. Species richness and diversity generally decreased across the studied plots. Longleaf pine tree density and importance value percent increased during the period. However, longleaf basal area ratio and aboveground biomass ratio in the stands decreased on average by 5% during the period, although these ratios increased in some locations in southwest Georgia and near the west coast of Florida. The longleaf pine basal area ratio and aboveground biomass ratio decreased equally in 37%, and increased in 19% and 21% of the study area, respectively. There was about 79% variation in the ratio of longleaf pine basal area among plots. When compared to the natural control of no disturbance, fire disturbance was significantly associated with greater longleaf pine basal area ratio and basal area growth. Understanding the change in growth and distribution patterns of longleaf pine across its range over time is vital to restore these critical ecosystems.

**Data Availability Statement:** The datasets used in this study are freely available to download on the website of the United States Department of Agriculture, Forest Service, Forest Inventory and

Analysis (FIA) DataMart (URL: https://apps.fs.usda.gov/fia/datamart). Any interested researcher can replicate this study by directly downloading data from the FIA website or contacting the Southern Research Station (SRS), FIA Research Work Unit, Knoxville for assistance in downloading the required data and variables.

**Funding:** This work was supported by the Agriculture and Food Research Initiative (AFRI) - Education and Workforce Development, grant no. 2019-67012-29700/project accession no.number 1019406, from the United States Department of Agriculture National Institute of Food and Agriculture (SKO and LDD). We acknowledge the support for publication from "Modeling the Impacts of Climate Change, Population Growth, and Land Use Change on Water Availability in Tennessee River Basin" USDA-NIFA Evans-Allen Project, Accession #number 1008993 (WT). The funders had no role in study design, data collection and analysis, decision to publish, or preparation of the manuscript

**Competing interests:** The authors have declared that no competing interests exist.

## Introduction

Longleaf pine (*Pinus palustris* Mill.) is an ecologically and economically valuable, and a high-priority conservation tree species of the southeastern United States [1]. The natural range of longleaf pine stretches along most of the Atlantic and Gulf Coastal Plains, from southeastern Virginia to eastern Texas [2]. The longleaf pine ecosystems once occupied around 24–36 million hectares of the South, of which roughly 1.3 million hectares remained by 2006, including about 4,856 hectares of fragmented old-growth forests with a high-diversity understory [1, 3, 4]. There are about 135 types of longleaf pine association communities, and longleaf pine forests are among the most diverse forest ecosystems in the US [1, 5]. Typically, longleaf pine ecosystems consist of widely spaced overstory trees over a predominantly herbaceous ground layer of grasses and forbs. The varied micro-environmental conditions across longleaf pine ecosystems make sites favorable for a great diversity of flora and fauna. Longleaf pine ecosystems provide refuge for at least 122 endangered or threatened plant species, and they give shelter to about 60 percent of the amphibian and reptile species in the southeast, many of them being endemic [4].

Several factors are responsible for the long-term decline of longleaf pine forests, including conversion of forests for cropland or pasture, harvesting without regenerating back to longleaf pine, fire exclusion, and reforestation of cutover areas with loblolly pine (*Pinus taeda* L.) and slash pine (*Pinus elliottii* Engelm.) [2, 3]. As a result of their range, longleaf pine forests can be subject to various disturbances, including tropical storms, blowdown, fire, lightning, and hydrological extremes [2]. The mortality of longleaf pine trees occurs due to a variety of disturbance agents, but weather (e.g., hurricanes, tornadoes, and lightning) is the primary cause of annual mortality of longleaf pine in the East Coastal Plain [1]. Mortality in longleaf pine stands can sometimes exceed increment, e.g., the average annual stand-level basal area increment per acre of longleaf pine (diameter 2.54 cm and greater) decreased by 3% to 31% in naturally regenerated stands in Alabama and Georgia, between 1972–1982 compared to growth rates in the same stands during the previous 10-year inventory cycle in each state [6]. Although the factors responsible for mortality that led to this decrease in longleaf pine growth were unknown, drought was identified as a possible factor for growth differences between two periods.

Surprisingly, fire can also be a major source of mortality: while fire is widely used as a silvicultural tool in ecological restoration of longleaf pine stands, the reintroduction of fire after a long period of suppression in longleaf pine stands with excessive fuel loading in the Flomaton Natural Area in Escambia County, Alabama killed a large portion of young longleaf pine roots between 1995 and 2003 and decreased longleaf pine density [7]. An earlier study [8] evaluated growth and mortality over 15 years in permanent plots established in 1999 in two old-growth stands in northern Alabama and reported a substantial tree mortality and initial decrease in longleaf pine basal area in response to periodic fires, although by 2014 both stands exceeded the levels of basal area measured in 1999.

The vulnerability of coastal ecosystems of the Southeast has increased in recent years because of the impacts of climate change through changes in hydrology, disturbance regimes, and interspecies interactions [9]. The wet longleaf pine savanna and Flatwoods of the Central Atlantic Coastal Plain is one of the most sensitive ecosystems to climate change in North America [10]. Many large-scale efforts for conservation and restoration of existing longleaf pine stands are currently underway, especially on federal and state lands, including on several national forests [3]. America's Longleaf Restoration Initiative (ALRI) is a range-wide effort between the public and private sectors to maintain, restore and conserve longleaf pine ecosystems across the South [11]. The Regional Conservation Partnership Program (RCPP) of the USDA has identified the longleaf pine forest ecosystem as a regional Critical Conservation

Area (CCA) to address natural resource conservation goals with improved agricultural productivity. Forest managers and landowners are in need of information about longleaf pine stand dynamics, long-term productivity, and competitive status of longleaf pine that could support restoration and active management in the future [1].

The irregularity in seed production by longleaf pine is a major cause of the poor natural regeneration of the species [12, 13]. Longleaf pine masts every seven to ten years and a profusion of cones are necessary for successful longleaf pine regeneration [14]. A declining proportion of longleaf pine in the overstory reduces its seed availability and natural regeneration occurrence in mixed forests [15]. It is still a matter of investigation whether the distributional pattern and compositional dominance of longleaf pine changed across the longleaf pine range during the last two decades. It is vital to know the change in proportion and density of longleaf pine across the longleaf range over a time period, as well as the variability of such a change. An understanding of the change in species composition and dominance and the relative growth rate of longleaf pine across the longleaf pine ecosystem over time is critical for sustainable management and conservation. Previous work [16] predicted that there could be either no growth or reduced growth in southern US forests as a result of the impact of the disturbances caused by climate change over the next century. Therefore, knowledge of the spatiotemporal dynamics of longleaf pine ecosystems will be helpful in the context of changing climate and disturbance regimes to plan and implement restoration and management strategies.

The Forest Inventory and Analysis (FIA) program of the United States Department of Agriculture (USDA) periodically samples across the United States to assess forest conditions. The FIA data is the primary source of information about the status and trends of US forest resources [17]. The FIA data has been widely used to study longleaf pine ecosystems, such as for mapping, habitat change and migration, stand structure and condition, and damages and disturbances [1, 18–20]. However, the changes in dominance and proportion of the composition of longleaf pine across the entire longleaf pine range over the last two decades have not yet been fully explored. Using the large and extensive periodic FIA dataset from the range of longleaf pine is vital to account for the change and variability in the structural and compositional status of longleaf pine, such as basal area ratio, basal area growth, and aboveground biomass growth during a period.

The main objective of our study was to investigate the change and variability in the proportion, density, growth, and dominance of longleaf pine across the longleaf pine associated forest types for the 1997–2018 period. Specifically, the objectives were to: i) investigate the spatial distribution of the changes in basal area, aboveground biomass, and species diversity for all species, ii) examine the spatial distribution of the change in longleaf basal area and basal area ratio, iii) test if the longleaf pine basal area ratio and basal area growth vary across the study region during the period, and iv) model the effect of disturbances that occurred on the plots at the beginning of the 1997–2018 period, on longleaf pine basal area ratio and basal area growth over the period. The study is original as it uses an extensive regional dataset to examine the magnitude of the spatial change in stand characteristics of longleaf pine associated forest types. The results will be useful for monitoring drivers of change in longleaf pine ecosystems and strengthening ongoing longleaf pine restoration and management strategies across the region.

## Methods and tools

### Study area

The study was conducted across longleaf pine associated forest types in the southeastern US, which included the states of Alabama, Florida, Georgia, Louisiana, Mississippi, North Carolina, South Carolina, and Texas (Fig 1). In the study area, landforms vary markedly across the

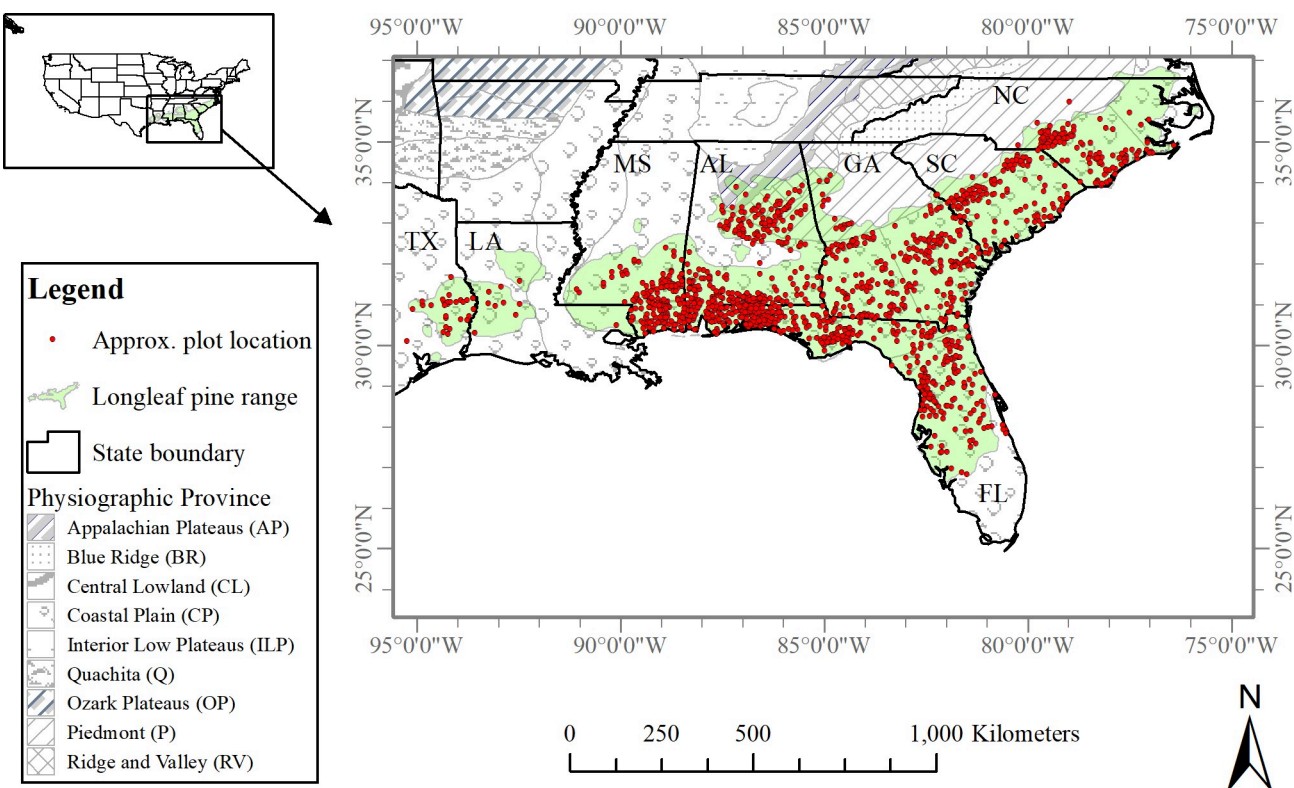

**Fig 1. The study area was the range of longleaf pine in the states of Alabama (AL), Florida (FL), Georgia (GA), Louisiana (LA), Mississippi (MS), North Carolina (NC), South Carolina (SC) and Texas (TX).** (Shapefile data sources: US state boundary [25], Physiographic Province [21], and Longleaf pine range [26]).

physiographic provinces of the Coastal Plain (Atlantic and Eastern Gulf), Piedmont, Appalachian and Cumberland Plateaus, and Ridge and Valley [21]. The terrain of the area differs from dissected irregular plains and high hills to rolling and mountainous landscapes, and from gently sloping hills to highly folded sandstone and limestone formations. Soil properties vary significantly across the region, but in general, deep and fine-textured soil, with clay or loamy subsoil, are common [22]. The southeastern US region has a humid subtropical climate characterized by mild winters, hot summers, and abundant annual precipitation [23]. This region is also rich in plant diversity as it comprises many forest types and consists of both mixed and pure forests, such as upland hardwood forests, planted pine, naturally regenerated pines, bottomland hardwoods, and oak-pine forests [24].

## Data and variables

We used forest inventory data from the USDA FIA, which is publicly available at the FIA Data-Mart (https://apps.fs.usda.gov/fia/datamart/). The FIA applies a nationally standardized sampling design with a systematic hexagonal cell grid, using a spatial sampling intensity of one plot per every approximately 2,400-hectare hexagonal cell [27]. The current national standard fixed-plot design was initiated by the FIA in 1995 and is used since then in all inventories. Data is collected on more than 300 variables at the level of the tree, stand, site, and ownership, and these variables can be used to report annual and periodic changes in the forests [28]. The FIA plot design consists of a cluster of four subplots, each of 7.3 m radius, where three circular subplots are placed 36.6 m apart, with their centers forming a triangular form, and a fourth

subplot is at the center of the triangle. Trees of diameter at breast height (dbh, 1.37 m above the ground) ≥12.7 cm are measured in the subplots. Within each subplot is nested a 2.07 m radius microplot to measure seedlings and saplings (dbh <12.7 cm). Details on the FIA plot design, layout, and measurement are described in the *FIA Database user guide* [29].

We selected two set of measurements of 1,432 plots collected during the inventory intervals 1997–2010 and 2003–2018, respectively. The 1,432 plots were not all measured at exactly the same time, so the first measurement of these plots occurred either in 1997 (when the new fixed-plot design started) or some time thereafter, but no later than 2010. The last measurement of these same plots occurred either in 2003 or later, but no later than in 2018. The overlap in the two periods, 1997–2010 and 2003–2018, means that some of the plots were getting their latest measurement, while others have not yet been measured for the first time. For simplicity, the first (1997–2010) and second (2003–2018) inventory intervals are referred hereafter by their centroids as '2004' and '2011', respectively, denoting the approximate mid-year of each inventory interval. All the sample plots were distributed within the current longleaf pine range [26] and contained longleaf pine in any one or more size categories, i.e., seedling, sapling, and tree. We used the plots if they had at least 10 percent canopy cover by live trees of any size and species, meeting the FIA's definition of forest land. The number of plots were 311 (21.7%) in Alabama, 406 (28.3%) in Florida, 252 (17.6%) in Georgia, 9 (0.6%) in Louisiana, 169 (11.8%) in Mississippi, 124 (8.7%) in North Carolina, 133 (9.3%) in South Carolina, and 28 (2.0%) in Texas (Fig 1).

The FIA plots are assigned to one or more forest types based on the tree species or species groups, forming the plurality of all live stocking. For the 2004 mid-year inventory period, the sample plots were classified into several forest type groups that included longleaf–slash pine (40.4% of the plots), loblolly–shortleaf pine (21.9%), oak-pine (17.8%), oak-hickory (16.5%) and other (3.4%, Fig 2A). The recorded maximum stand age was 117 years for the plots of 2004 (Fig 2B).

Data variables were broadly categorized into two groups: stand structure and disturbance condition. Stand structure variables are derived from individual tree level data. We used the

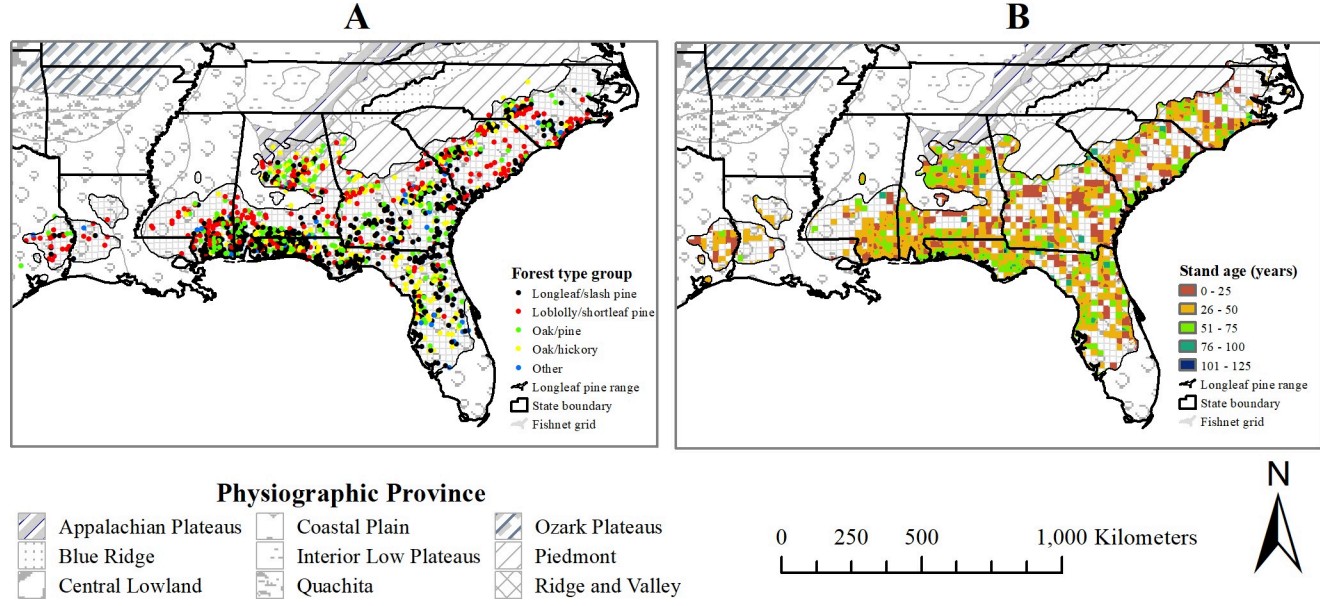

**Fig 2.** Approximate spatial distribution of the A) plots with associated dominant forest type groups and B) stand age classes in the 2004 dataset (first inventory interval 1997–2010). (Shapefile data sources: US state boundary [25], Physiographic Province [21], and Longleaf pine range [26]).

variables species name, dbh, and height of each tree. As per FIA classification, *tree* refers to all stems of dbh ≥12.7 cm in the midstory and overstory of the stand, and we only used tree data in this study. Other computed variables were quadratic mean diameter, total aboveground biomass (Mg ha$^{-1}$), tree density per hectare, total basal area (m$^2$ ha$^{-1}$), longleaf pine basal area (m$^2$ ha$^{-1}$), longleaf pine aboveground biomass (Mg ha$^{-1}$), longleaf pine basal area ratio to total basal area, and longleaf pine aboveground biomass ratio to total aboveground biomass, basal area coefficient of variation of longleaf pine trees (%), and aboveground biomass coefficient of variation of longleaf pine trees (%). Aboveground biomass ratio and basal area ratio of longleaf pine value range from 0 to 1, representing a pure longleaf pine stand at a value of 1. Species richness and Shannon's diversity index were used as diversity indices. Shannon's diversity index was calculated based on the basal area rather than on the number of trees because the basal area accounts for size variation [30].

The FIA records disturbance on the plots since the previous measurement, for periodically measured plots, or within the last five years for new plots. As per the FIA definition, a disturbance is any event that caused damage or mortality to at least 25 percent of the trees and affected an area of at least 0.4 hectares. Many plots had damage caused by multiple disturbance agents. The gap between the first and second measurement was more than one inventory cycle for a large number of selected plots. Therefore, we included disturbances (single or compound) only from the first measurement (2004) as a baseline of each plot. The disturbance condition of the plots is referred hereafter as the 'plot condition' and is broadly classified into seven groups (Table 1). Silvicultural treatments included site preparation, fertilizing, girdling, pruning, and herbicide treatment. The 'plot condition' was a categorical variable of seven classes.

## Methods of analysis

We calculated the aboveground dry biomass (AGB) of each live tree using published biomass equations [31] and adjustment factors developed by the FIA for the tree components [29]. Descriptive statistics, such as mean, standard deviation, and minimum and maximum values, were computed for stand structural attributes for 2004 and 2011. The dependent paired-samples t-test was applied to compare the means of 2004 and 2011 of each stand variable to determine whether there was a statistically significant difference between these means at α = 0.05 level.

**Table 1. Description of plot condition classes based on impact by single or multiple disturbances across the plots (N = 1,432).**

| Plot condition code | Plot (%) | Description |
|---|---|---|
| ND | 54.7 | No disturbance |
| C | 21.4 | Plots had harvesting (cutting) and silvicultural treatments |
| F | 11.0 | Plot disturbance predominantly by fire, but also affected by secondary and tertiary disturbance agents, such as insect, disease, animal, weather or human |
| FC | 3.6 | Plot disturbance predominantly by fire, but also affected by secondary and tertiary disturbance agents, such as insect, disease, animal, weather or human, and also affected by harvesting and silvicultural treatments |
| W | 3.6 | Plot disturbance predominantly by weather, but also affected by secondary and tertiary disturbance agents, such as animal, fire or human |
| WC | 2.1 | Plot disturbance predominantly by weather, but also affected by secondary and tertiary disturbance agents, such as animal, fire or human, and had harvesting and silvicultural treatments |
| OTH | 3.6 | Plot disturbance predominantly by other agents, such as animals, diseases, insects, human, other vegetation and unknown, but also had harvesting and silvicultural treatments |

The percent of live, dead, and removed trees on the plots were calculated for 2004 and 2011. Diameter class of live longleaf pine trees, as well as of all the trees across the plots, were analyzed for 2004 and 2011 to observe the diameter distribution status during the period.

We calculated the importance value percent (IVP) of the tree species for 2004 and 2011. The IVP of each species was the average of relative frequency percent, relative density percent, and relative basal area dominance percent [32]. The use of IVP is widespread in ecological studies because it is not influenced by large trees or a large number of small trees from a particular species [33].

A descending ranking of species for tree density, basal area, and IVP for 2004 and 2011 was done to identify the dominance order of the species across the plots. From it, we identified ten species that exhibited the highest increase and ten species that showed the highest decrease in their tree density, basal area, and IVP between 2004 and 2011.

We used the Inverse Distance Weighted (IDW) spatial interpolation to visualize and predict the spatial variability of forest variables [34] based on the data from the studied plots. IDW estimates cell values by averaging the values of sample data points in the neighborhood of each processing cell. The nearer points from the center of the cell being estimated have a greater influence or weight in the averaging process than do the farther points. We overlaid the study area with a fishnet grid of rectangular cells (0.2˚ × 0.2˚) to aggregate the point data, normalize geography for mapping and produce output in regularly shaped polygon features. Fishnet serves as a framework for raster data sets, in which each cell is assigned a value for the spatial variables of interest. The ArcMap default search radius of 12 data points was used for interpolation. The georeferenced data points of each plot for the variables basal area, aboveground biomass, longleaf pine basal area ratio, longleaf pine aboveground biomass ratio, coefficient of variation of basal area of longleaf pine trees, coefficient of variation of aboveground biomass of longleaf pine trees, stand age, species richness, and Shannon's diversity index were interpolated using the IDW estimate to produce raster data images for 2004 and 2011. Then, the change detection raster image was produced to detect the spatial locations of change and estimate the magnitude of change of the selected stand variables across the study area over the period. The change detection raster images were grouped using a manual method classifier. The five class breaks were added by setting five appropriate class ranges of change vectors, showing increasing and decreasing trend relative to 0. The five classes of the data were recognized as 'Greatly decreased,' 'Decreased,' 'No change,' 'Increased,' and 'Greatly increased.' The second level class interval of either 'Increased' or 'Decreased' was an approximate mid-class taken as the midpoint of the range between 'No change' and the greatest value of the change vector (but smaller one of two negatively greatest and positively greatest values). The area of each class was calculated based on pixel counts and cell size (in hectares) in the raster image. The relative area covered by a class was the percentage of the total study area of the five classes.

We used linear mixed modeling to explore how basal area or basal area ratio of longleaf pine changed across the plots throughout the period 1997–2018. Our sample plots were not remeasured at an equal time interval but at different time points. Therefore, we assumed that all the plots had the first measurement at baseline time 0 and the second measurement at different times from 1.8 to 22.7 years. We then applied growth models that do not require the time points to be equally spaced out across observations and can be used with unbalanced and missing data [35]. The time (remeasurement years) was nested within plots to examine the growth of basal area and basal area ratio across 2004 (first measurement) through 2011 (last measurement) and the effect that plot condition (disturbances) has on basal area and basal area ratio. The Satterthwaite approximation for the degrees of freedom was used in the mixed model to address the problem of the unequal sample sizes (different number of plots in the different plot conditions) and variance heterogeneity of the groups [36]. We applied the model

building process [35], to estimate multilevel models and to obtain the best fitting and parsimonious model for the sampled dataset (Table 2).

All four models were estimated using maximum likelihood (ML), and model-fit was evaluated using Akaike's Information Criterion (AIC) and Bayesian Information Criterion (BIC) when the nested models differed in either random or fixed effects. At first, an unconditional model with no predictors was applied to calculate the intraclass correlation coefficient (ICC) to assess between-plot variation in longleaf pine basal area or basal area ratio (model 1, Table 2). ICC is the ratio of the intercept variance to total variance (intercept + residual). It also reveals the average longleaf pine basal area or basal area ratio for 2004 and if these variables in 2004 varied among plots. The linear model building process for model 2 to 4, which included fixed and random effects along with model output, are described in Table 2. While time and plot condition as fixed-effect predictors measure the effect of time and plot condition on longleaf basal area or basal area ratio across all plots, time as a random-effect measures the variance in the effect of time on longleaf pine basal area or basal area ratio across plots.

A general linear model where the variable time is the only predictor in a simple level-1 equation can be defined as [35]:

$$Y_{ij} = \beta_{0j} + \beta_{1j} * X_{ij} + e_{ij} \qquad (1)$$

Where $Y_{ij}$ is the longleaf pine basal area or basal area ratio at $i^{th}$ time in $j^{th}$ plot. $\beta_{0j}$ is the average longleaf pine basal area or basal area ratio for $j^{th}$ plot. $X_{ij}$ is a $i_{th}$ time predictor in $j_{th}$ plot, and $\beta_{1j}$ is the slope or regression coefficient associated with $X_{ij}$, where this value shows the relationship between the time and longleaf pine basal area or basal area ratio, $e_{ij}$ is the time-level error term that is assumed to be normally distributed with covariance R.

The simple level-2 model when time is nested in plots for random-effect is expressed in equations as [35]:

$$\beta_{0j} = \gamma_{00} + \gamma_{01} * W_j + \mu_{0j} \qquad (2)$$

$$\beta_{1j} = \gamma_{10} + \mu_{1j} \qquad (3)$$

Where $\gamma_{00}$ is the intercept representing the grand mean of longleaf pine basal area or basal area ratio across time and plots, $W_j$ is predictor time for $j^{th}$ plot, and $\gamma_{01}$ estimates the average effect of time. By adding $\mu_{1j}$ error term in the level-2 Eq (3), we make the relationship between time ($X_{ij}$) and the longleaf basal area or basal area ratio ($Y_{ij}$) to vary across the plots. The level-2 errors are assumed normally distributed with covariance G.

**Table 2. Linear models building process.**

| model 1 | model 2 | model 3 | model 4 |
|---|---|---|---|
| Unconditional model<br>No predictors<br>Included just random-effect for the intercept | model 1 + predictor 'time' as fixed-effect | model 2 + random slopes for predictor 'time' | model 3 + predictor 'plot condition' as fixed-effect |
| Output used to calculate Intraclass Correlation Coefficient (ICC) that gives information on how much variation in the 'longleaf pine basal area' or 'basal area ratio' exists between the plots | Results show the relationship between 'time' and longleaf pine 'basal area' or 'basal area ratio' | Fixed-effect results the same information as model 2.<br><br>In addition, random slope results indicate if the relationships between 'time' and the longleaf pine 'basal area or basal area ratio' vary among plots. | In addition to the results provided by the model 3, output also reveals if there is a relationship between 'plot condition' and longleaf pine 'basal area' or 'basal area ratio'. |

A final combined level-1 and level-2 random intercept and slope model (Eq 4) is structured by merging Eqs (1), (2) and (3) into model 3:

$$\gamma_{ij} = \gamma_{00} + \gamma_{01} * W_j + \gamma_{10} * X_{ij} + \mu_{0j} + \mu_{1j} * X_{ij} + e_{ij} \tag{4}$$

The model estimates a continuous longleaf pine basal area or basal area ratio outcome ($Y_{ij}$), an intercept ($Y_{00}$), level-1, and level-2 regression coefficients ($Y_{10}$ and $Y_{01}$, respectively), and level-1 and level-2 error terms ($e_{ij}$ and ($\mu_{0j}$, $\mu_{1j}$), respectively).

The statistical packages IBM SPSS 25 and SAS 9.4 were used for data analysis and statistical inferences. SAS PROC MIXED procedure with multiplicity adjustment was used for two-level growth modeling. Species richness, Shannon's diversity index, and Importance Value Percent (IVP) were calculated using PC-ORD Version 6.12. Spatial data analysis and map production were performed using ArcMap 10.6.1.

## Results

We found a significant change in the stand structure and species composition across the plots between 2004 (centroid-year of first inventory interval 1997–2010) and 2011(centroid-year of last inventory interval 2003–2018), as described in more detail below.

### Structural changes

**Changes in stand characteristics.**   We found greater quadratic mean diameter, mean height, mean aboveground biomass per hectare, and mean basal area per hectare of the live trees in 2011 than in 2004 (Table 3). Likewise, tree density, basal area, and aboveground biomass per hectare significantly increased for longleaf pine in 2011. However, the proportion of longleaf pine aboveground biomass and proportion of its basal area significantly decreased during the period (Table 3). The average tree species diversity of the plots significantly decreased during the period: the average species richness and Shannon's diversity index on the plots were 4.4 and 0.92, respectively in 2004, and 4.3 and 0.89, respectively, in 2011 (Table 3). The plots had a similar level of variation of longleaf pine aboveground biomass and basal area around their mean in 2004 and 2011, with coefficient of variation 58% and 46%, respectively, in 2004, and 57% and 47%, respectively in 2011. The average plot remeasurement period (time) was 12.5 years, with a minimum of 1.8 years to a maximum of 22.7 years.

**Diameter size distribution.**   The plots had a high percent of trees in the lowest three diameter size classes: about 95% of the trees in 2004 and 94% in 2011 had a diameter between 12.7 and 40 cm (Fig 3A). Likewise, about 99% of longleaf pine trees had a diameter between 12.7 and 50 cm in 2004 and in 2011 (Fig 3B).

**Percent dead, live, and removed trees.**   The percent of live trees declined from 95.7% to 77.3% between 2004 and 2011 (Fig 4A). The percent of live longleaf pine trees also decreased from 96.7% in 2004 to 83.3% in 2011 (Fig 4B). Harvesting and silvicultural treatments were one of the predominant causes of tree removal for longleaf and other species across the plots by the end of the period.

### Species composition change

Species richness across the plots was 84 in 2004 and 79 in 2011, which was a statistically significant decline (Table 3). Based on the IVP ranking of the species, the most dominant four species were longleaf pine, loblolly pine, slash pine and water oak (*Quercus nigra* L.) in both 2004 and 2011 (S1 Table). While the increase in basal area and IVP was greatest for loblolly pine, the increase in tree density was greatest for longleaf pine during the period (Fig 5A–5C). Loblolly

**Table 3. Structural changes of all species and longleaf pine across the plots by 2004 and 2011.** Where 2004 and 2011 are the centroid-year of first (1997–2010) and last (2003–2018) inventory intervals, respectively. Each pair of variables were tested using the dependent paired-samples t-test at α = 0.05 level (N = 1,432).

| Variables | Centroid-year of inventory interval | Plots | Mean | Std. Dev. | Min | Max | t-statistics |
|---|---|---|---|---|---|---|---|
| **All species** | | | | | | | |
| **Quadratic mean diameter (QMD, cm)** | 2004 | 1,396 | 23.9 | 5.7 | 12.7 | 59.4 | $(t_{1,395} = -9.1, p<0.0001)$ |
| | 2011 | 1,396 | 25.1 | 6.3 | 12.7 | 60.6 | |
| **Mean height (m)** | 2004 | 1,396 | 15.9 | 4.0 | 5.2 | 34.7 | $(t_{1,395} = -14.9, p<0.0001)$ |
| | 2011 | 1,396 | 17.2 | 4.0 | 6.4 | 33.5 | |
| **Tree density (stems ha$^{-1}$)** | 2004 | 1,396 | 276 | 177.2 | 15 | 1249 | $(t_{1395} = -1.6, p = 0.1)$ |
| | 2011 | 1,396 | 284 | 181.5 | 15 | 1026 | |
| **Basal area (m$^2$ ha$^{-1}$)** | 2004 | 1,396 | 12.3 | 8.2 | 0.2 | 48.8 | $(t_{1,395} = -6.2, p<0.0001)$ |
| | 2011 | 1,396 | 13.6 | 8.6 | 0.2 | 49.2 | |
| **Aboveground biomass (Mg ha$^{-1}$)** | 2004 | 1,396 | 63.6 | 51.3 | 0.2 | 346.9 | $(t_{1,395} = -9.3, p<0.0001)$ |
| | 2011 | 1,396 | 74.2 | 55.9 | 0.1 | 341.6 | |
| **Species richness** | 2004 | 1,396 | 4.4 | 2.6 | 1 | 14 | $(t_{1,395} = 2, p = 0.04)$ |
| | 2011 | 1,396 | 4.3 | 2.6 | 1 | 15 | |
| **Shannon's diversity index** | 2004 | 1,396 | 0.92 | 0.6 | 0 | 2.26 | $(t_{1,395} = 2.5, p = 0.01)$ |
| | 2011 | 1,396 | 0.89 | 0.6 | 0 | 2.32 | |
| **Longleaf pine** | | | | | | | |
| **Tree density (stems ha$^{-1}$)** | 2004 | 1,396 | 70 | 89.4 | 0 | 714 | $(t_{1,395} = -2.4, p = 0.017)$ |
| | 2011 | 1,396 | 76 | 106.1 | 0 | 788 | |
| **Basal area (m$^2$ ha$^{-1}$)** | 2004 | 1,396 | 3.9 | 4.8 | 0 | 36.8 | $(t_{1,395} = -4.2, p<0.0001)$ |
| | 2011 | 1,396 | 4.3 | 5.3 | 0 | 39.1 | |
| **Basal area ratio** | 2004 | 1,396 | 0.36 | 0.3 | 0 | 1 | $(t_{1,395} = 3.2, p = 0.001)$ |
| | 2011 | 1,396 | 0.34 | 0.3 | 0 | 1 | |
| **Aboveground biomass (Mg ha$^{-1}$)** | 2004 | 1,396 | 22.6 | 31.2 | 0 | 277.7 | $(t_{1,395} = -5.8, p<0.0001)$ |
| | 2011 | 1,396 | 25.8 | 35.5 | 0 | 296.3 | |
| **Aboveground biomass ratio** | 2004 | 1,396 | 0.38 | 0.3 | 0 | 1 | $(t_{1,395} = 3.6, p = 0.0003)$ |
| | 2011 | 1,396 | 0.36 | 0.3 | 0 | 1 | |
| **Basal area coefficient of variation (%)** | 2004 | 712 | 46 | 24.2 | 0 | 134 | $(t_{711} = -1.37, p = 0.17)$ |
| | 2011 | 712 | 47 | 24.5 | 0 | 144 | |
| **Aboveground biomass coefficient of variation (%)** | 2004 | 712 | 58 | 29.1 | 0 | 183 | $(t_{711} = 0.45, p = 0.65)$ |
| | 2011 | 712 | 57 | 30.2 | 0.4 | 206 | |

Note: Longleaf pine basal area and aboveground biomass coefficient of variation were estimated only for those plots that had more than one longleaf pine trees on the plot in both 2004 and 2011. Some pair of plots were excluded because of absence of trees of any species in those plots either in 2004 or 2011.

pine was the second most dominant tree species with an abundance of trees in the large diameter-size classes. While slash pine tree density and IVP declined the most, shortleaf pine (*Pinus echinata* Mill.) had the greatest decline in basal area among all species. The hardwood species that exhibited the greatest increase in tree density and IVP was water oak, and the hardwood species with the greatest growth in basal area was laurel oak (*Quercus laurifolia* Michx.) (Fig 5A–5C). On the other hand, turkey oak (*Quercus laevis* Walter) had the greatest decrease in tree density, basal area, and IVP among the hardwood species across the plots during the period.

## Spatial distribution of the change in total basal area, total aboveground biomass, and species diversity

Change detection images (Fig 6) showed variability in the distribution of the change in total basal area, total aboveground biomass, and species diversity across the study area during the

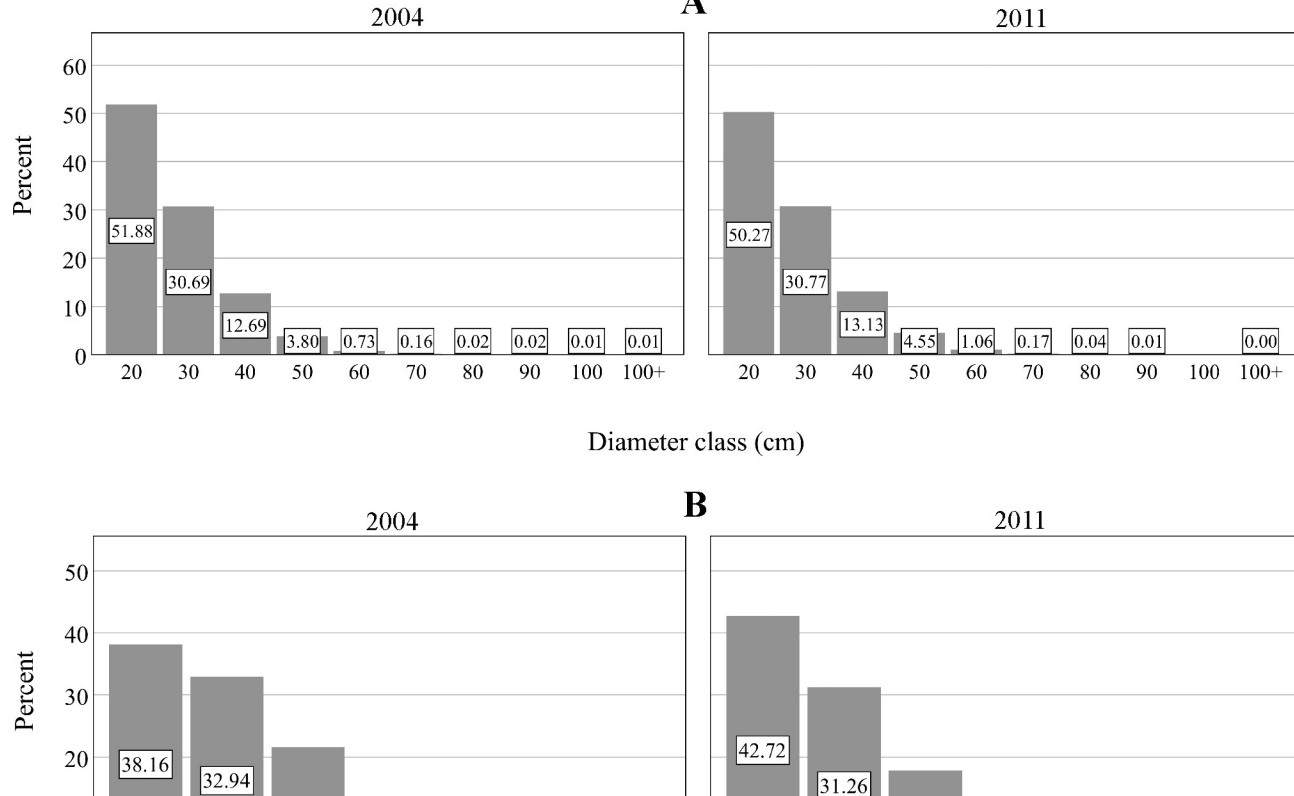

**Fig 3.** Diameter class distribution of live trees for (A) all species and (B) longleaf pine in 2004 and 2011 across the plots, where 2004 and 2011 are the centroid-year of first (1997–2010) and last (2003–2018) inventory intervals, respectively.

period. There was no change in total basal area in about 5% of the study area, but it increased in 62%, and increased substantially in 2% of the study area. The basal area decreased in about 25% and greatly decreased in 5% of the study area (Fig 6A and Table 4). Similarly, the percent of the area that showed no change, increased, and greatly increased aboveground biomass was 1%, 69%, and 3%, respectively (Fig 6B and Table 4). While basal area and aboveground biomass apparently increased in the Gulf Coastal Plain south of Mississippi, Alabama, and adjoining northwest Florida, both greatly increased in some areas of the Piedmont and Ridge and Valley along central Alabama and southeast Georgia and in North Carolina. The change detection map (Fig 6) showed an increase in total basal area and total aboveground biomass across the study area during the period. The species richness did not change in 34% of the study area, but it decreased in 31% (1 to 4 species), and greatly decreased in 3% (5 to 13 species) of the area (Fig 6C and Table 4). Species richness increased by 1 to 4 species in 31% and highly increased by 5 to 9 species in 1% of the area. Similarly, the area with decreased species diversity (40%) was almost equal to the area with increased species diversity (39%) (Fig 6D and Table 4). Only 2% of the area had highly increased species diversity (Shannon's diversity index increased by 0.9 to 1.7). Species richness and diversity increased greatly in some plots across the Piedmont and Ridge and Valley in Alabama and southeast Georgia.

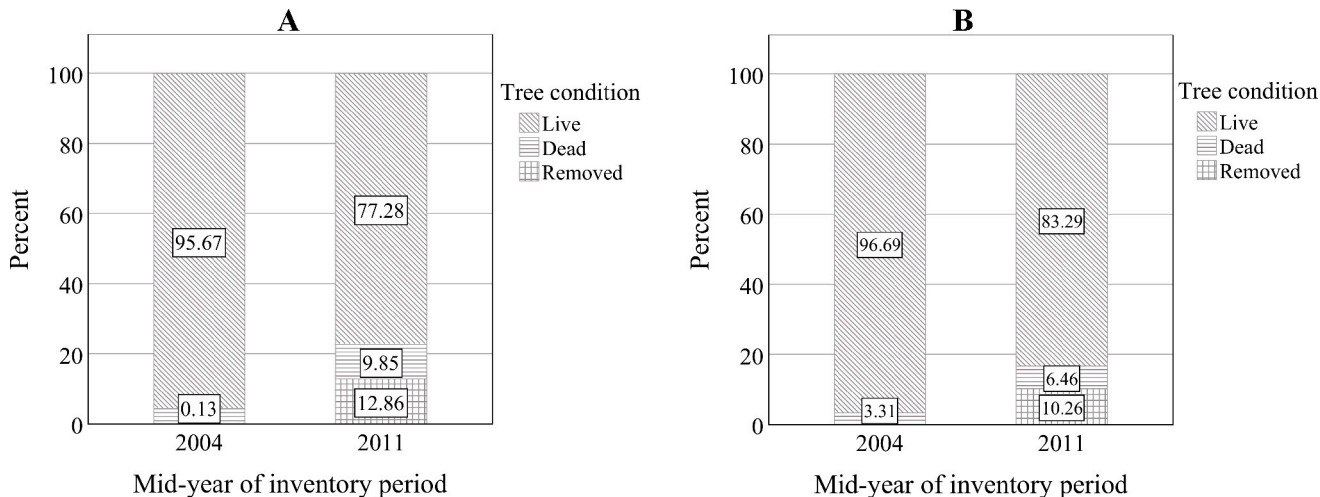

**Fig 4.** Percent of live, dead, and removed trees for (A) all species and (B) longleaf pine across the plots in 2004 and 2011, where 2004 and 2011 are the centroid-year of first (1997–2010) and last (2003–2018) inventory intervals, respectively.

## Spatial distribution of changes in relative basal area and relative aboveground biomass of longleaf pine

According to the change detection map (Fig 7A), the basal area ratio of longleaf pine decreased in 35% and greatly decreased in 2% of the study area. While about 44% of the study area had no change in longleaf pine basal area ratio for 2004 and 2011, 19% of the area had an increase in that ratio (Fig 7A and Table 5). Similarly, the aboveground biomass ratio of longleaf pine decreased in 37% and increased in 21% of the study area (Fig 7B and Table 5). Overall, the basal area ratio and aboveground biomass ratio of longleaf pine decreased by a little over 5% between 2004 and 2011. Longleaf pine basal area ratio and aboveground biomass ratio increased, however, in some areas of southern Mississippi, southwest Georgia and near the West Coast of Florida. There was no change in the basal area coefficient of variation of longleaf pine trees in only 2% of the total study area between 2004 and 2011 (Fig 7C and Table 5). Longleaf pine basal area coefficient of variation decreased in 51% of the study area, which was slightly higher than the area where it increased (47%). The coefficient of variation of the longleaf pine aboveground biomass decreased in 55% and increased in 42% of the study area during the period (Fig 7D and Table 5). The increase in the basal area coefficient of variation or aboveground biomass coefficient of variation of longleaf pine trees was substantial in some of the plots (stand age <50 years) of southwest Georgia.

## Variation in basal area ratio and basal area growth of longleaf pine over a 22-year period

In both cases of basal area ratio and basal area of longleaf pine, the AIC and BIC values decreased from model 1 to model 4, indicating better model fit throughout the progression of models with a most parsimonious and best-fitting model 4 (S2 and S3 Tables). In the unconditional growth model 1 (S2 Table), the Intraclass Correlation Coefficient (ICC) of 0.79 for the response variable longleaf pine basal area ratio suggested that there was 79% variation in the longleaf pine basal area ratio among the plots. In the best model 4 (S2 Table), both time and plot condition (disturbances) were statistically significant predictors for longleaf pine basal

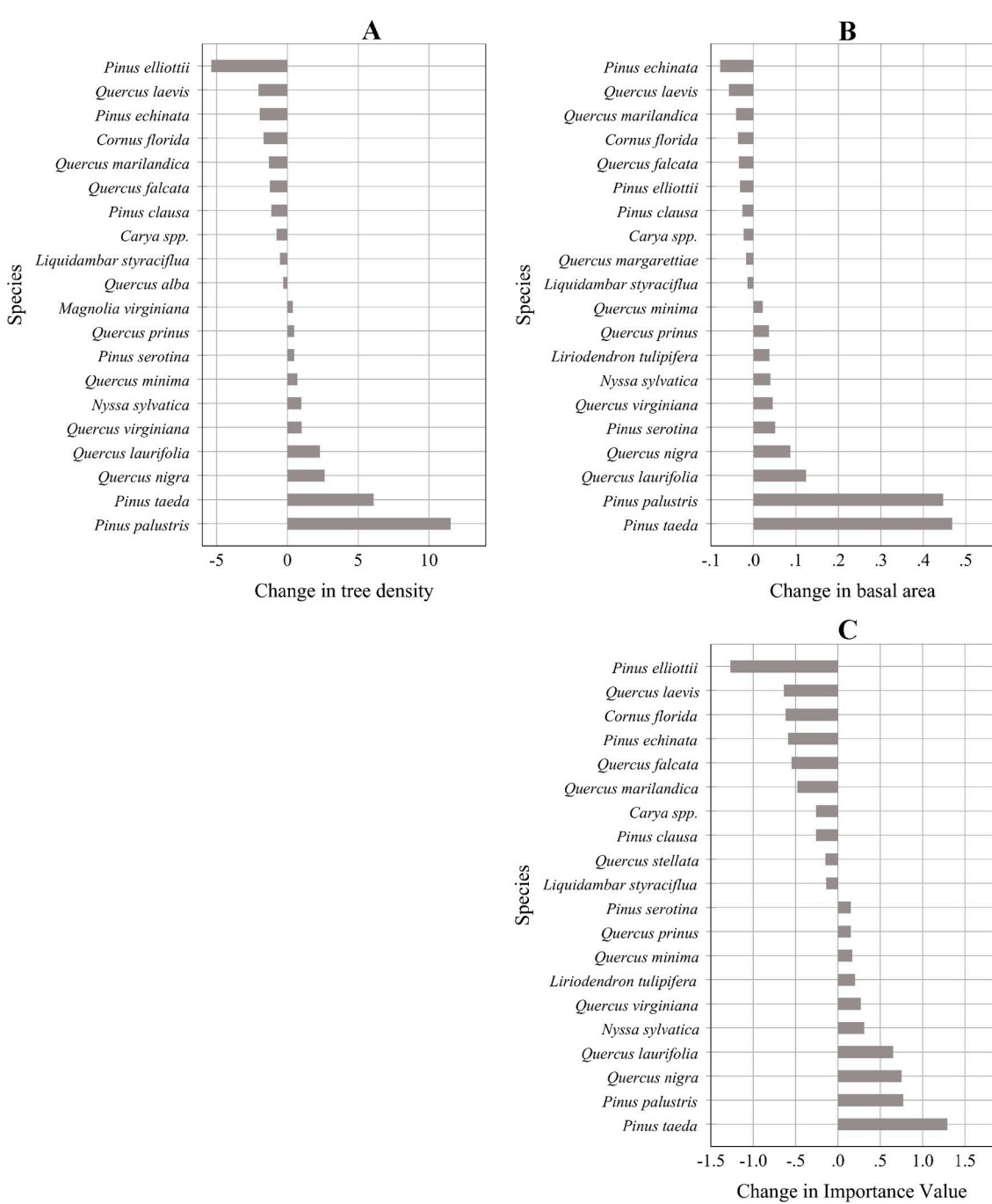

**Fig 5.** Ten species with the greatest increase and ten species with the greatest decrease in (A) tree density (stems ha⁻¹), (B) basal area (m² ha⁻¹), and (C) Importance Value (%) across the plots between 2004 and 2011, where 2004 and 2011 are the centroid-year of first (1997–2010) and last (2003–2018) inventory intervals, respectively.

area ratio. For every one-year change, on average, the longleaf pine basal area ratio of the plots decreased by 0.1% ($p = 0.008$). The plots that experienced fire disturbance had significantly greater growth ($b = 0.16$, $p < 0.001$), while those experiencing weather disturbance had significantly smaller growth ($b = -0.09$, $p = 0.04$) in longleaf pine basal area ratio compared to plots with no disturbance.

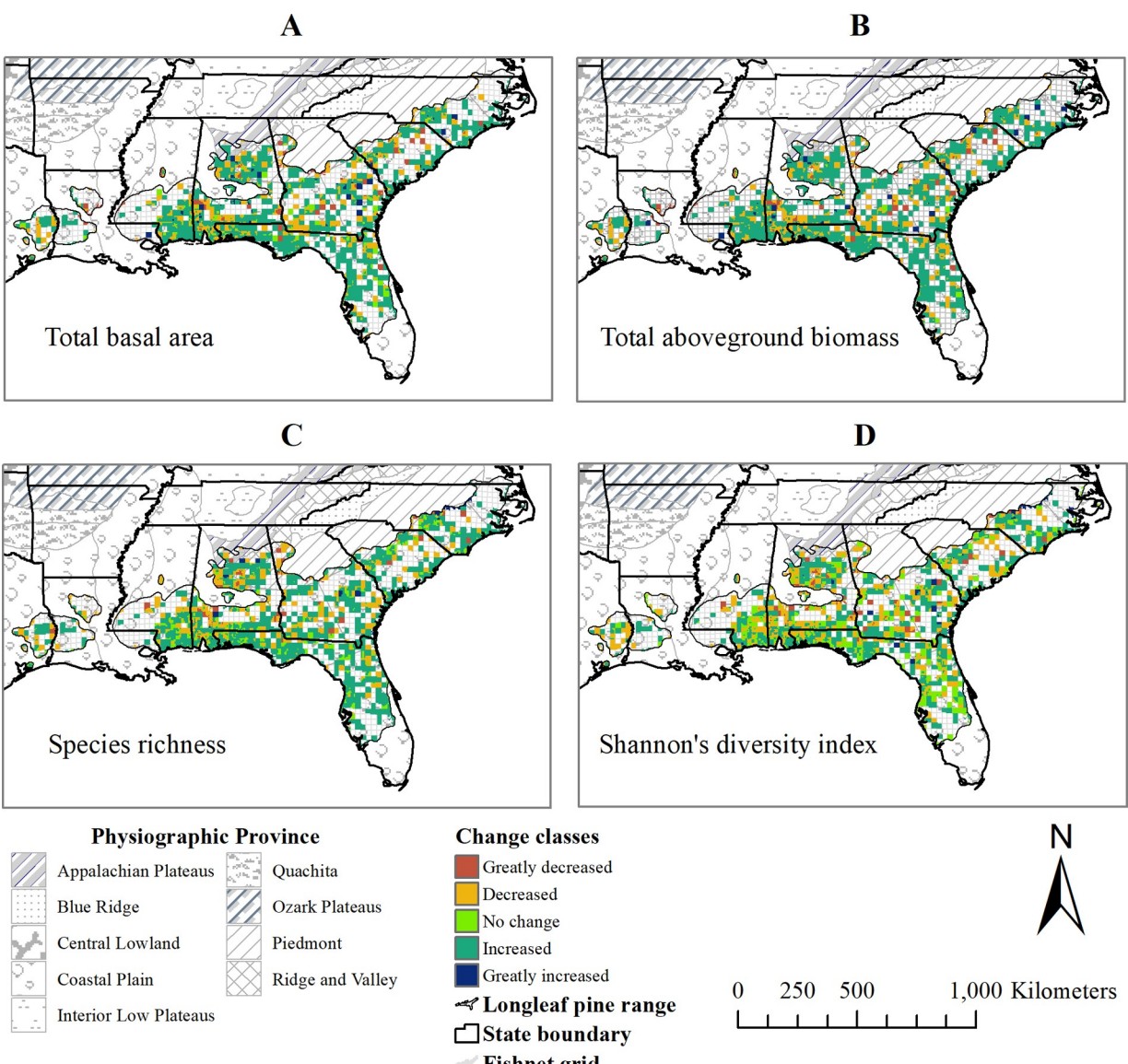

**Fig 6.** Change detection maps showing the spatial distribution of change in (A) total basal area (m² ha⁻¹), (B) total aboveground biomass (Mg ha⁻¹), (C) species richness, and (D) Shannon's diversity index, in the study area between 2004 and 2011. Where 2004 and 2011 are the centroid-year of first (1997–2010) and last (2003–2018) inventory intervals, respectively. (Shapefile data sources: US state boundary [25], Physiographic Province [21], and Longleaf pine range [26]).

Similar to the longleaf pine basal area ratio, the Intraclass Correlation Coefficient (ICC) of 0.78 for longleaf pine basal area in the unconditional growth model 1 (S3 Table) suggested that there was 78% variation in longleaf pine basal area among the plots. According to the best model 4, longleaf pine basal area of the plots increased by 3% ($p < 0.001$) per year on average (S3 Table). The statistically significant random-effect of time indicates that longleaf pine basal area across the plots changed at different rates during the 22 years we examined. Plots with harvesting and silvicultural treatments had smaller ($b = -1.1$, $p = 0.0004$), and plots with fire had greater ($b = 2.2$, $p < 0.001$) longleaf pine basal area growth when compared to no disturbance plots (S3 Table).

The longleaf pine basal area ratio and aboveground biomass ratio both decreased over the period (Fig 8A and 8B), while the longleaf pine basal area, longleaf pine aboveground biomass,

**Table 4. Change classes for total basal area, total aboveground biomass, and species diversity across the plots between 2004 and 2011, where 2004 and 2011 are the centroid-year of first (1997–2010) and last (2003–2018) inventory intervals, respectively (N = 1,432).**

| Change in | Classes of change vector | Total study area based on pixel cell size (ha) | Relative study area coverage | Change classes |
|---|---|---|---|---|
| **Total basal area (m² ha⁻¹)** | -35 to -14 | 1,656,323.4 | 5% | Greatly decreased |
| | -13 to -1 | 7,693,745.0 | 25% | Decreased |
| | 0 | 1,631,735.6 | 5% | No change |
| | 1 to 13 | 18,910,250.6 | 62% | Increased |
| | 14 to 27 | 681,752.5 | 2% | Greatly increased |
| **Total aboveground biomass (Mg ha⁻¹)** | -253 to -76 | 1,414,915.9 | 5% | Greatly decreased |
| | -75 to -1 | 7,029,874.5 | 23% | Decreased |
| | 0 | 283,877.3 | 1% | No change |
| | 1 to 75 | 20,964,449.2 | 69% | Increased |
| | 76 to 155 | 880,690.2 | 3% | Greatly increased |
| **Species richness** | -13 to -5 | 1,095,274.6 | 3% | Greatly decreased |
| | -4 to -1 | 9,665,239.2 | 31% | Decreased |
| | 0 | 10,619,692.7 | 34% | No change |
| | 1 to 4 | 9,752,414.1 | 31% | Increased |
| | 5 to 9 | 447,050.8 | 1% | Greatly increased |
| **Shannon's diversity index** | -2.3 to -0.9 | 1,410,445.4 | 4% | Greatly decreased |
| | -0.8 to -0.1 | 11,442,266.3 | 36% | Decreased |
| | 0 | 6,500,119.2 | 21% | No change |
| | 0.1 to 0.8 | 11,632,262.9 | 37% | Increased |
| | 0.9 to 1.7 | 594,577.6 | 2% | Greatly increased |

total aboveground biomass and total basal area all increased (Fig 8C–8F). The y-axis (linear predictor) of each figure displays the scale of the outcome variable (Fig 8).

## Discussion

### Changes in density, proportion, and spatial distribution of longleaf pine

Among the major factors that contributed to the long-term decline of longleaf pine ecosystem in the southeastern region of the US are: large-scale logging without regeneration in the early 1900s, fire exclusion, conversion of forests for cropland or pasture, lack of seed trees for natural regeneration, poor survival of planted seedlings due to competition, and reforestation of cutover areas with loblolly and slash pine [2, 3, 37]. Using the FIA data, an earlier study [1] found a declining geographic extent and density of longleaf pine between 1970 and 2010. In our analysis of FIA data, we found a decline in the proportion of longleaf pine in some locations in southern Alabama, south and central Georgia, and eastern parts of Florida between 2004 (centroid-year of first inventory interval 1997–2010) and 2011 (centroid-year of last inventory interval 2003–2018). While tree density, basal area, and importance value percent of longleaf pine increased across the longleaf associated forest ecosystems (S1 Table), the proportion of longleaf pine basal area and aboveground biomass, relative to the total stand basal area and biomass, has decreased significantly (on an average around 5%) over the period. There were 30 species whose basal area increased by a greater percentage than did the basal area of longleaf pine. The decrease in longleaf basal area ratio was associated with a greater increase in the basal area ratio particularly of (all increased at rates faster than longleaf, and are listed here in order of the greatest amount of absolute basal area increase) loblolly pine and some hardwood species, such as laurel oak, water oak, black cherry (*Prunus serotina* Ehrh.), southern live oak (*Quercus virginiana* Mill.), blackgum (*Nyssa sylvatica* Marshall), yellow poplar (*Liriodendron tulipifera* L.) and dwarf live oak

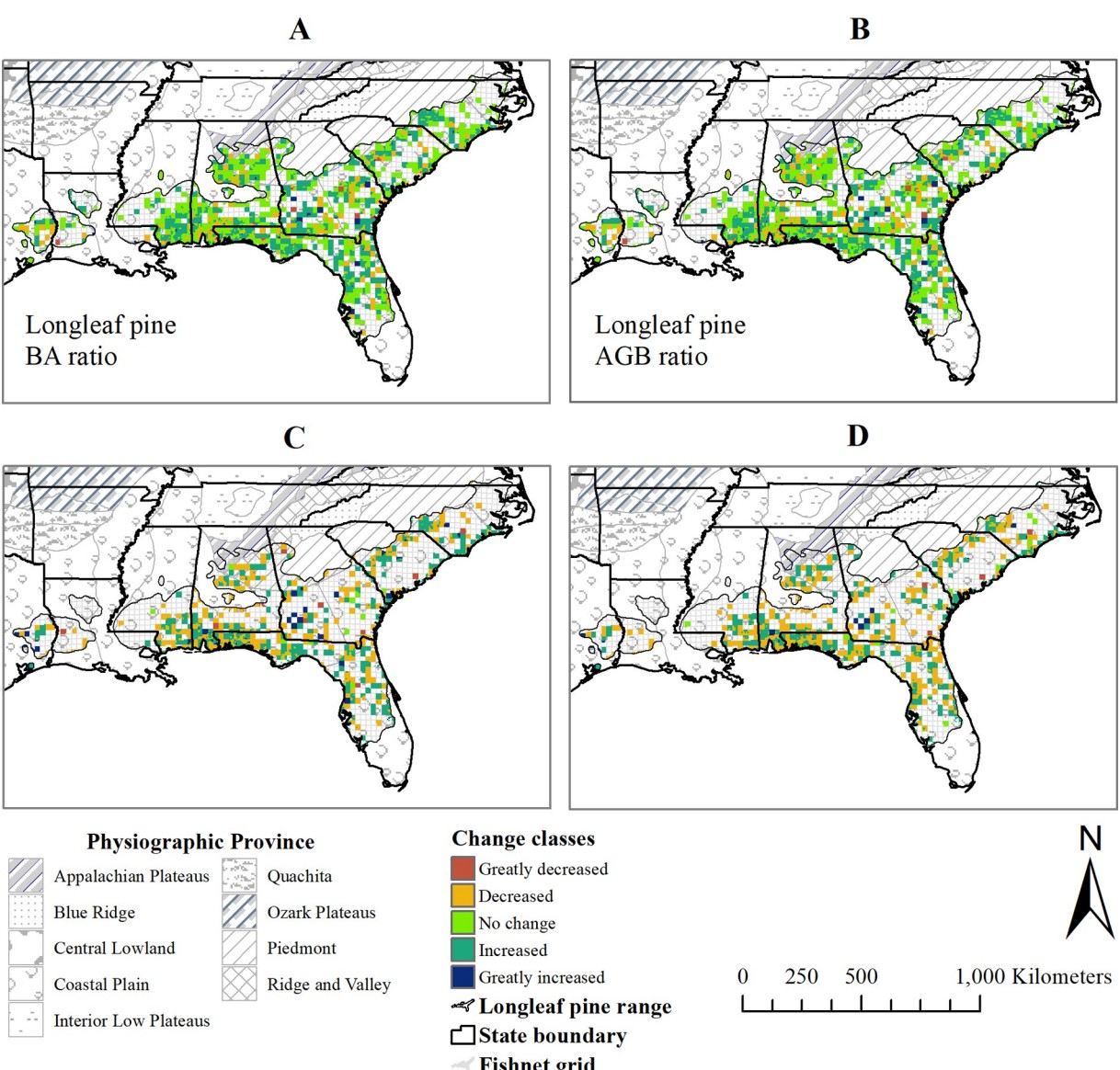

**Fig 7.** Change detection maps showing the spatial distribution of changes in longleaf pine (A) basal area (BA) ratio, (B) aboveground biomass (AGB) ratio, (C) basal area coefficient of variation, and (D) aboveground biomass coefficient of variation, in the study area between 2004 and 2011, where 2004 and 2011 are the centroid-year of first (1997–2010) and last (2003–2018) inventory intervals, respectively. (Shapefile data sources: US state boundary [25], Physiographic Province [21], and Longleaf pine range [26]).

(*Quercus minima* (Sarg.) Small) (S1 Table). Slash pine decreased substantially during the period, and possibly that favored other species more than it favored longleaf pine (Fig 5). Due to the reduction of fire frequency during the early and mid-20th century in longleaf pine forests, hardwoods increased in the midstory and overstory layers with a dominance of shrubs in the understory [38]. Fire suppression causes the development of a scrub oak midstory in many existing xeric and subxeric sandhills longleaf pine forests. An earlier study [39] reported a shift in stand structure, favoring succession and high density of hardwoods because of the long-term exclusion of fire in old-growth longleaf pine forests in the Sandhills of North Carolina. In those old-growth longleaf pine forests, while black oak (*Quercus velutina* Lam.), hickories (*Carya* spp.), and large, sparse longleaf pines were dominant woody species on mesic sites, turkey oak and

**Table 5. Change classes for basal area ratio, aboveground biomass ratio, basal area coefficient of variation and aboveground biomass coefficient of variation of longleaf pine across the plots between 2004 and 2011, where 2004 and 2011 are the centroid-year of first (1997–2010) and last (2003–2018) inventory intervals, respectively (N = 1,432).**

| Change in | Classes of change vector | Total study area based on pixel cell size (ha) | Relative study area coverage | Change classes |
|---|---|---|---|---|
| Longleaf pine basal area ratio | -1 to -0.6 | 558,813.6 | 2% | Greatly decreased |
| | -0.5 to -0.1 | 10,814,159.9 | 35% | Decreased |
| | 0 | 13,690,932.0 | 44% | No change |
| | 0.1 to 0.5 | 5,606,017.6 | 18% | Increased |
| | 0.6 to 1 | 386,699.0 | 1% | Greatly increased |
| Longleaf pine aboveground biomass ratio | -1 to -0.6 | 578,930.8 | 2% | Greatly decreased |
| | -0.5 to -0.1 | 10,979,568.7 | 35% | Decreased |
| | 0 | 12,982,356.4 | 42% | No change |
| | 0.1 to 0.5 | 6,111,185.0 | 20% | Increased |
| | 0.6 to 1 | 404,581.0 | 1% | Greatly increased |
| Longleaf pine basal area coefficient of variation (%) | -77 to -39 | 505,328.3 | 3% | Greatly decreased |
| | -38 to -1 | 9,232,302.8 | 48% | Decreased |
| | 0 | 476,260.7 | 2% | No change |
| | 1 to 38 | 8,145,623.4 | 42% | Increased |
| | 39 to 101 | 983,825.0 | 5% | Greatly increased |
| Longleaf pine aboveground biomass coefficient of variation (%) | -143 to -72 | 207,944.8 | 1% | Greatly decreased |
| | -71 to -1 | 10,526,927.0 | 54% | Decreased |
| | 0 | 391,294.0 | 2% | No change |
| | 1 to 71 | 7,812,464.5 | 40% | Increased |
| | 72 to 156 | 404,709.8 | 2% | Greatly increased |

the longleaf pines were abundant on xeric sites. Flatwoods understories often get dominated by many fire resistant species, such as saw-palmetto (*Serenoa repens* Bartram) J.K.Small), gallberry (*Ilex glabra* (L.) Gray), waxmyrtle (*Myrica cerifera* L.), and sweetgum (*Liquidambar styraciflua* L.) [40]. Only repeated fires at short return intervals over a long period can significantly control these woody species [41].

The change in the coefficient of variation in the basal area or aboveground biomass of longleaf pine trees in some of the plots was possibly due to the impact of disturbances on recruitment, growth, and mortality of the species in those plots.

Nearly all longleaf pine trees (about 99%) had diameter between 12.7 and 50 cm, and a small portion (1%) were >50 cm in both 2004 and 2011, with a reverse-J diameter class distribution (Fig 3). These younger cohorts are valuable for the development of a range of vertical structures and vital for the long-term sustainability of the longleaf pine and associated ecosystems. The stands of older age classes in longleaf pine-dominated stands in some locations in the South have adequate advance regeneration because of the common unevenaged structure of these forest [1]. This high variability (78–79%) in longleaf pine basal area and basal area ratio among the plots also indicated that longleaf pine stands had a high vertical structural diversity or size inequality across the study area.

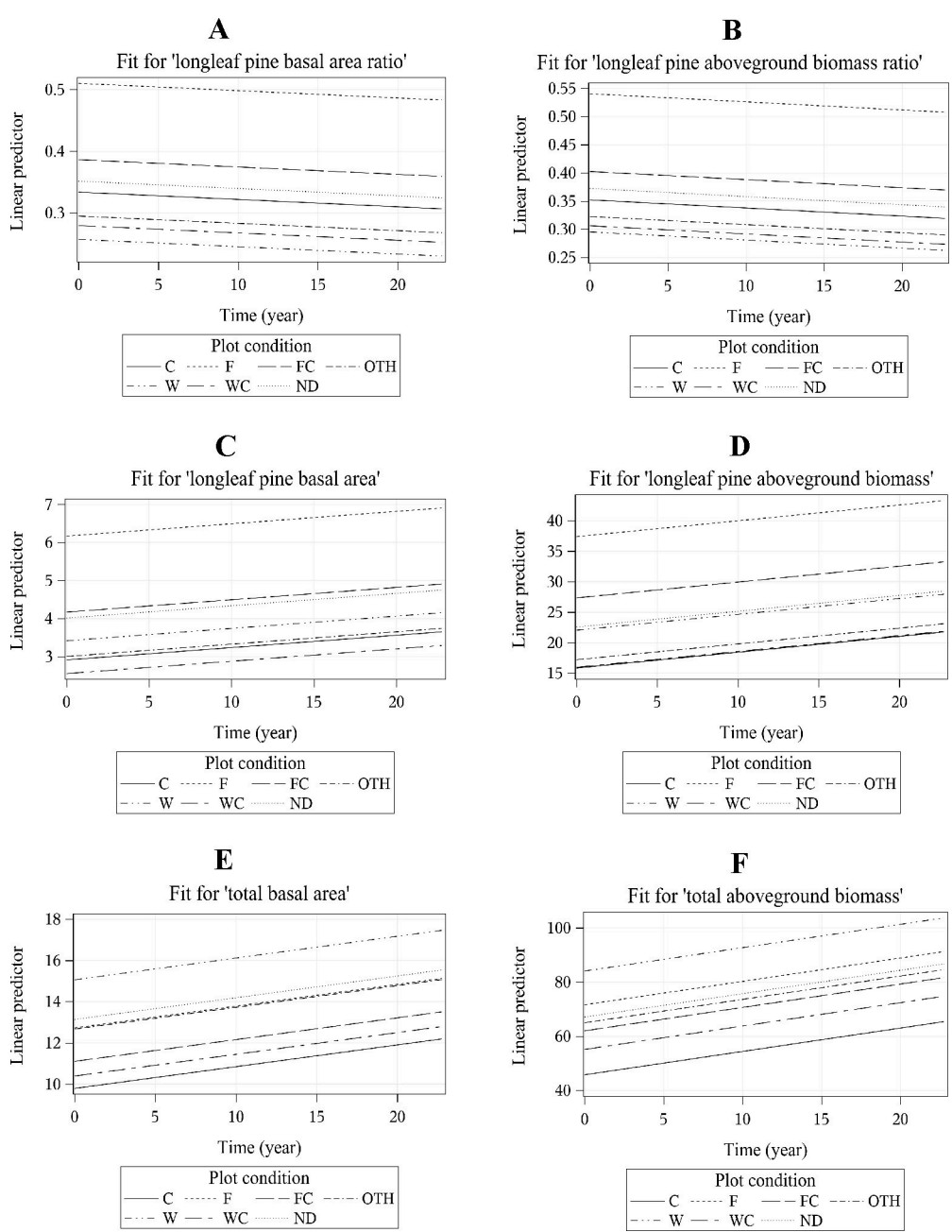

**Fig 8. Fitted effect plots of linear mixed models (model 4) exhibit the relationship between outcome variables and plot remeasurement period (time) by plot condition.** Basal area measured in m$^2$ ha$^{-1}$ and aboveground biomass in Mg ha$^{-1}$. Plot condition codes are described in Table 1.

Compared to other locations, we found a greater increase in total basal area and total aboveground biomass in the panhandle of Florida, southern Alabama, southern Mississippi, in the Piedmont and the Ridge and Valley along central Alabama and in southeast Georgia. Species composition in southeastern longleaf pine forests has changed substantially between 1970 and 2010 due to the conversion of large portion of longleaf pine acreage to the loblolly pine and longleaf pine-oak forest types [1]. The decrease in species richness and diversity across the stands during the period that we studied was likely because of the removal of many hardwood

species from the midstory and overstory, such as bitternut hickory (*Carya cordiformis* (Wangenh.) K. Koch), elm (*Ulmus spp.*), shagbark hickory (*Carya ovata* (Mill.) K. Koch), chinquapin oak (*Quercus muehlenbergii* Engelm.) willow (*Salix spp.*), pecan (*Carya illinoinensis* (Wangenh.) K. Koch), mulberry (*Morus spp.*), eastern white pine (*Pinus strobus* L.), and bastard oak (*Quercus sinuata* Walter) during harvesting or silvicultural treatments or mortality in the period (S1 Table), because these species were absent in the plots during the second measurement. However, there were also recruitments of hardwood species, such as Ogeechee tupelo (*Nyssa ogeche* W. Bartram ex Marshall), silktree (*Albizia julibrissin* Durazz.) and nutmeg hickory (*Carya myristiciformis*, (Michx. f.) Nutt.) in the midstory in 2011 that were not found in 2004.

## Disturbances and longleaf pine growth

Disturbances occur on average in about 5% of the area of longleaf pine-dominated forests in the South annually, with the Piedmont area having the lowest rate (3%), and the west Coastal Plain having the highest (6%) annual disturbance rate [1]. We found fire, weather, harvesting, and silvicultural treatments as significant and predominant factors regarding the growth of longleaf pine. When compared to no disturbance, fire had a positive and weather, harvesting, and silvicultural treatment had a negative association with the longleaf pine basal area and basal area ratio. Both natural and anthropogenic fires are the most important drivers of stand dynamics of southern pine ecosystems, where it is frequently applied as a management tool [3, 20, 39]. The fire recurrence in many southern pine forests is generally less than ten years [42]. Lightnings are a major cause of forest fires in southern pine forests, accounting for about 18% of the fires and 35% of the area burned in Florida [43]. However, the frequency of occurrence of prescribed and lightning-started fires, in general, substantially decreases in drought years [44].

An earlier study [1] reported disturbances caused by fire occurring on about 85% and by weather on 7% of the 79,000 hectares of longleaf pine forests where a disturbance was recorded. Longleaf pine stands can recover and exceed their pre-fire stand basal area rapidly. Although fire initially results in massive mortality of small size stems of longleaf pine in old-growth longleaf pine woodlands, it does not create a persistent negative impact on the long-term density of that size class because of the strong recovery of small trees [8]. Frequent surface fires favor longleaf pine establishment by preventing the regeneration and growth of competitive but less fire-tolerant species [38]. Three or four prescribed fires applied at every 2 or 3 years produced positive changes in the overstory, midstory and understory layers of longleaf communities in the Coastal Plain of Alabama and are considered useful for longleaf pine restoration [38].

Weather-related disturbance followed by salvage cutting was likely the reason for a substantial reduction in longleaf pine stocking on weather impacted plots. Disturbance from weather (mainly hurricanes and windstorms) frequently occurs along southern coastal areas and are likely significant drivers of stand dynamics in these ecosystems [20, 39, 45]. Many intense hurricanes, including Ivan (2004), Katrina (2005), Rita (2005), and Gustav (2008) have struck southern coastal states during the last decade [46]. Substantial damage by wind disturbance occurs in large-diameter stands of pines and hardwoods [20]. However, longleaf pine is relatively more resistant to wind damage than loblolly pine, possibly due to stem strength, rooting habit, and canopy characteristics [37].

Silvicultural activities such as planting, thinning, or fertilization are commonly applied treatments across longleaf pine-dominated forests. We found harvesting and silvicultural treatments as a significant factor associated with the reduction of longleaf pine growth. Similarly, a previous work [1] observed harvesting to be the most common treatment, occurring annually on 40% of

the longleaf pine forests that experienced anthropogenic disturbances or treatments. Longleaf pine harvests mostly occur on privately owned properties, rather than public land [1].

## Management implications

Understanding the changes in species composition and their growth rates is essential to the management and restoration of forests. Knowledge of the changes in the spatial distribution of growth patterns of longleaf pine across its range can be useful for identifying critical locations and implementing restoration activities.

The mixed longleaf pine or longleaf pine-oak stands that primarily contain abundant hardwoods in their midstory could quickly be restored to a functioning longleaf pine system with minimal management intervention [1]. The most competing hardwood species can be reduced by treatments such as injection with herbicides, girdling, or harvesting to increase the availability of moisture, nutrients, and light for the regeneration in longleaf stands [47]. Therefore, the reduction of competing hardwood species, including the frequently occurring water oak, laurel oak, sweetgum, southern live oak, blackgum, and red maple (*Acer rubrum* L.) (which were abundant in the last measurements of the studied plots) can be useful to enhance the natural regeneration and growth of longleaf pine. When fires become less frequent, hardwood succession begins in small patches and invades longleaf woodlands over time [1]. This ecosystem alteration intensifies as hardwood trees become mature and fire-resistant, increasing the availability of hardwood seed sources and quantity of less pyrogenic fuels on site. The use of frequent fire with silvicultural treatments that mimic intermediate-severity natural disturbances, such as group selection and irregular shelterwood, can reduce hardwoods and develop a conducive environment for recruiting and establishing adequate natural regeneration and younger cohorts of longleaf pine [48]. The longleaf pine cone crops vary greatly by year and location, which is a major problem for the natural regeneration of this species [13, 49]. We found longleaf density to have decreased substantially in some locations of Georgia and South Carolina in 2011 (as per the last inventory for those particular plots). Therefore, an adequate number of seed trees should be maintained in those stands to increase the natural regeneration of longleaf pine. An earlier study [15] found that a one-unit increase in the overstory basal area ratio of longleaf pine likely increases the odds of occurrence of their seedlings by approximately seven times in oak-pine forest type groups of the southeast US.

## Conclusion

The long-term decline of longleaf pine from southern forests has been a growing concern among forest managers, silviculturists, and conservationists for many years. Several governmental and non-governmental organizations have collaborated to increase the presence of this important pine species throughout the southern forests. We found increased tree density, basal area, and importance value percent of longleaf pine across its range during 1997–2018. However, the ratio of longleaf pine basal area to the total stand basal area and the ratio of its aboveground biomass to total stand biomass have both declined, on average, around 5% over the same period. When compared to no disturbance, fire had a positive and weather, harvesting and silvicultural treatments had a negative association with longleaf pine basal area ratio and basal area growth. The presence of a range of stand structures with abundant younger longleaf cohorts across longleaf forests will possibly increase the long-term sustainability of this species and associated ecosystems. Maintaining structural heterogeneity of longleaf pine stands could also increase their resilience to disturbance events. The preservation of healthy longleaf pine ecosystems will be essential for safeguarding long-term economic benefits to landowners, develop crucial wildlife habitat, and enhance ecosystem vitality in the southern

landscape. This study only selected plots with longleaf pine present in any one or more size categories, i.e., seedling, sapling, and tree from the longleaf pine range. Therefore, the estimated importance value percent or dominance ranking of longleaf pine applies only to the plots sampled and not to the full range that it may inhabit.

## Supporting information

**S1 Table. Stem density, basal area, and importance value percent sorted by species in 2004 and 2011.** The highest-ranking values are in bold. All 1,432 plots were included in this analysis, including 36 plots that had no trees (dbh$\geq$ 12.7 cm) either in 2004 or 2011. Where 2004 and 2011 are the centroid-year of first (1997–2010) and last (2003–2018) inventory intervals, respectively.
(DOCX)

**S2 Table. Estimates from linear mixed models examining longleaf pine basal area ratio (N = 1,432).**
(DOCX)

**S3 Table. Estimates from linear mixed models examining longleaf pine basal area ($m^2$ $ha^{-1}$) (N = 1,432).**
(DOCX)

## Acknowledgments

We are grateful to the forest inventory crews who collected the data and to Bill Burkman, Jeff Turner and Jason McHan from the United States Department of Agriculture, Forest Service, Forest Inventory and Analysis (FIA), Knoxville, Tennessee, for their support in obtaining the data.

## Author Contributions

**Conceptualization:** Santosh K. Ojha, Luben D. Dimov.

**Data curation:** Santosh K. Ojha.

**Formal analysis:** Santosh K. Ojha.

**Funding acquisition:** Santosh K. Ojha, Luben D. Dimov, Wubishet Tadesse.

**Investigation:** Santosh K. Ojha.

**Methodology:** Santosh K. Ojha.

**Project administration:** Santosh K. Ojha, Wubishet Tadesse.

**Resources:** Luben D. Dimov, Wubishet Tadesse.

**Software:** Luben D. Dimov, Wubishet Tadesse.

**Supervision:** Luben D. Dimov, Wubishet Tadesse.

**Validation:** Santosh K. Ojha.

**Writing – original draft:** Santosh K. Ojha.

**Writing – review & editing:** Luben D. Dimov, Wubishet Tadesse.

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
