## [Editor Report · Decision Letter 0]

4 Nov 2020

PONE-D-20-19610

Growth, proportion, and distribution pattern of longleaf pine across southeastern forests and disturbance types: a change assessment for the period 1997-2018

PLOS ONE

Dear Dr. Ojha,

Thank you for submitting your manuscript to PLOS ONE. After careful consideration, we feel that it has merit but does not fully meet PLOS ONE’s publication criteria as it currently stands. Therefore, we invite you to submit a revised version of the manuscript that addresses the points raised during the review process.

We look forward to receiving your revised manuscript.

Kind regards,

Ines Ibáñez, Ph.D.

Academic Editor

PLOS ONE

Additional Editor Comments:

This is a descriptive study of changes taking place across longleaf pine ecosystems in SE USA. This information could be relevant for conservation of these systems and as guidance for future research projects aimed at identifying and quantifying agents of change. The study is sound and well written but needs a few clarifications.

Years used in the analysis should be specified clearly, title mentions 1997-2018, while analysis refers to 2004 and 2011 centroids of 1997-2010 and 2003-2018 intervals, make clear what it was used/done specially because the intervals overlap and it is not clear if plots were counted twice. My suggestion would be to use the time intervals, e.g., 1-22 years, rather than actual years. Also, the intervals between censuses was quite different 1 to 22 years, changes should be standardized per year basis (something done in the analysis but rate of change is likely not linear, e.g., much higher after disturbance).

Methods, L230 and beyond. It is not clear if the analysis is based on FAI data solely and the IDW was only for the purpose of visualizing results at sites not included in the analysis; or if there was a second analysis of remote sensing data. Please, clarify. If the latest, I can see interpolating biomass, but I’m not sure that would work for diversity, please, explain what it was done and its feasibility.

L420-422 what does actually mean?

Results: it is not clear what is the value of showing, and commenting on, all models. My suggestion is to only mention, and comment on, model with best fit (model 4), and have those tables moved to the supplement.

For all the variables and ratios measured indicated what changes on its values would indicate about those sites, e.g., natural successional dynamics, degradation, changes in community composition, …

Journal Requirements:

2. We note that Figures 1, 2, 6, 7 in your submission contain map images which may be copyrighted. All PLOS content is published under the Creative Commons Attribution License (CC BY 4.0), which means that the manuscript, images, and Supporting Information files will be freely available online, and any third party is permitted to access, download, copy, distribute, and use these materials in any way, even commercially, with proper attribution. For these reasons, we cannot publish previously copyrighted maps or satellite images created using proprietary data, such as Google software (Google Maps, Street View, and Earth). For more information, see our copyright guidelines: http://journals.plos.org/plosone/s/licenses-and-copyright.

2.1.    You may seek permission from the original copyright holder of Figures 1, 2, 6, 7 to publish the content specifically under the CC BY 4.0 license. 

2.2.    If you are unable to obtain permission from the original copyright holder to publish these figures under the CC BY 4.0 license or if the copyright holder’s requirements are incompatible with the CC BY 4.0 license, please either i) remove the figure or ii) supply a replacement figure that complies with the CC BY 4.0 license. Please check copyright information on all replacement figures and update the figure caption with source information. If applicable, please specify in the figure caption text when a figure is similar but not identical to the original image and is therefore for illustrative purposes only.

---

## [Author Response · Author response to Decision Letter 0]

30 Nov 2020

We have addressed all the comments and suggestions of the Academic Editor and reviewer. A file labeled 'Response to Reviewers' is uploaded along with other documents.

---

## [Editor Report · Decision Letter 1]

26 Dec 2020

Growth, proportion, and distribution pattern of longleaf pine across southeastern forests and disturbance types: a change assessment for the period 1997-2018

PONE-D-20-19610R1

Dear Dr. Ojha,

We’re pleased to inform you that your manuscript has been judged scientifically suitable for publication and will be formally accepted for publication once it meets all outstanding technical requirements.

Kind regards,

Ines Ibáñez, Ph.D.

Academic Editor

PLOS ONE

Additional Editor Comments (optional):

The authors have successfully addressed the comments brought up in the review.
---

## [Editor Report · Acceptance letter]

7 Jan 2021

PONE-D-20-19610R1 

Growth, proportion, and distribution pattern of longleaf pine across southeastern forests and disturbance types: a change assessment for the period 1997-2018 

Dear Dr. Ojha:

I'm pleased to inform you that your manuscript has been deemed suitable for publication in PLOS ONE. Congratulations! Your manuscript is now with our production department. 

Kind regards, 

on behalf of

Dr. Ines Ibáñez 

Academic Editor

PLOS ONE